# VELIE: A Vehicle-Based Efficient Low-Light Image Enhancement Method for Intelligent Vehicles

**DOI:** 10.3390/s24041345

**Published:** 2024-02-19

**Authors:** Linwei Ye, Dong Wang, Dongyi Yang, Zhiyuan Ma, Quan Zhang

**Affiliations:** 1Department of Electrical and Electronic Engineering, University of Liverpool, Liverpool L69 3BX, UK; linwei.ye21@student.xjtlu.edu.cn (L.Y.); dong.wang21@student.xjtlu.edu.cn (D.W.); dongyi.yang20@student.xjtlu.edu.cn (D.Y.); 2School of Advance Technology, Xi’an Jiaotong-Liverpool University, 111 Ren’ai Road, Suzhou 215123, China; 3Department of Electrical and Computer Engineering, The University of Texas at Austin, Austin, TX 78712, USA

**Keywords:** autonomous driving, low-light image enhancement, RGB sensor image processing

## Abstract

In Advanced Driving Assistance Systems (ADAS), Automated Driving Systems (ADS), and Driver Assistance Systems (DAS), RGB camera sensors are extensively utilized for object detection, semantic segmentation, and object tracking. Despite their popularity due to low costs, RGB cameras exhibit weak robustness in complex environments, particularly underperforming in low-light conditions, which raises a significant concern. To address these challenges, multi-sensor fusion systems or specialized low-light cameras have been proposed, but their high costs render them unsuitable for widespread deployment. On the other hand, improvements in post-processing algorithms offer a more economical and effective solution. However, current research in low-light image enhancement still shows substantial gaps in detail enhancement on nighttime driving datasets and is characterized by high deployment costs, failing to achieve real-time inference and edge deployment. Therefore, this paper leverages the Swin Vision Transformer combined with a gamma transformation integrated U-Net for the decoupled enhancement of initial low-light inputs, proposing a deep learning enhancement network named Vehicle-based Efficient Low-light Image Enhancement (VELIE). VELIE achieves state-of-the-art performance on various driving datasets with a processing time of only 0.19 s, significantly enhancing high-dimensional environmental perception tasks in low-light conditions.

## 1. Introduction

In Advanced Driving Assistance Systems (ADAS), Automatic Driving Systems (ADS), and Driver Assistance Systems (DAS), RGB camera sensors are extensively employed due to their low cost and high information density. RGB imaging in photos and videos plays a crucial role in various intelligent driving tasks such as depth estimation, object detection, semantic segmentation, object tracking, and 3D high-dimension mapping. Current research in intelligent driving vehicle perception systems has evolved into systems comprising pure RGB sensing, RGB-guided Lidar and Radar perception, and multi-sensor fusion systems incorporating RGB [1,2,3]. Favored for their low manufacturing and deployment costs, RGB cameras, however, face significant limitations in robustness under complex environments, especially in low-light conditions. Nighttime and low-illumination driving scenarios, unavoidable for vehicles, however, diminish human reaction and perception capabilities, both visually and cognitively, necessitating enhanced perception aids. The limited nighttime performance of RGB not only threatens its reliability in intelligent driving but also challenges the accuracy and safety of intelligent driving systems.

The fundamental reason for the poor nighttime performance of RGB cameras lies in their imaging principle. Relying on visible light for image capture, RGB imaging effectiveness is highly susceptible to external light sources. At nighttime, the absence of natural light sources like sunlight results in weak environmental perception, making it challenging for cameras to capture clear signals. The vehicle’s own light sources, such as headlights, cannot compensate for this global illumination deficiency. Additionally, the sensitivity of RGB camera sensors to light diminishes significantly under low-light conditions. This indicates that the signal-to-noise ratio of the camera sensor decreases under dim lighting, capturing weaker effective light signals relative to the noise generated by the sensor and circuits.

Beyond the deficits in image clarity and noise, low-light conditions also degrade the color capture capability of vehicle-mounted RGB cameras. RGB cameras capture colors through a tri-color filter array (red, green, blue), but in poorly lit environments, the sensor’s ability to capture these colors weakens, causing color distortion or insufficient color information. Furthermore, the dynamic range of nighttime environments, which indicates the measuring range from darkest to brightest areas, often exceeds the processing capabilities of standard vehicle-mounted RGB sensors, resulting in the loss of detailed information. Such high-frequency information, like the contours of objects, colors, ground markings, and traffic light signals, serve as essential recognition features for autonomous and assisted driving; the loss of this information in dark areas significantly increases the likelihood of accidents [4].

To counter the deficiencies of RGB cameras in darkness, past research approaches have been divided into two aspects: hardware improving development and software image signal processing. On the hardware side, the use of sensors like Lidar, event cameras, radar, and thermal [5,6,7,8], which do not primarily rely on illumination for signal acquisition, has been proposed to assist RGB in intelligent automotive tasks. However, the fusion of multiple sensors greatly diminishes the low-cost advantage of RGB cameras. Both in terms of deployment and sensor hardware costs, the cost of using multi-sensor fusion is significantly higher than the cost of multiple RGB cameras. On the other hand, professional low-light cameras produced by companies like Sony, Photonis, SiOnyx, and Texas Instruments have also emerged in the market, but due to their high prices, they are not suitable for deployment in driving scenarios. Professional low-light cameras [9,10] utilize high-performance Charge-Coupled Devices (CCD) or Complementary Metal-Oxide-Semiconductor (CMOS) technology, professional low-light circuits, and filters as core components to enhance low-light imaging quality; hence, their manufacturing process is highly stringent, technically complex, and very costly. Conversely, improving software algorithms offers a more cost-effective alternative, namely enhancing images through preprocessing of the original RGB images. Thus, some research preprocesses the raw data from RGB cameras and then utilizes these enhanced data for high-dimensional perception tasks. This task on low-light images is collectively referred to as **L**ow-**L**ight **I**mage **E**nhancement (LLIE).

Research on Low-Light Image Enhancement (LLIE) can primarily be categorized into three distinct approaches: traditional LLIE methods, deep learning-based LLIE methods, and LLIE methods tailored for driving scenarios. Each of these categories has been the subject of extensive investigation within their respective domains. Upon reviewing past research on LLIE tasks in driving scenarios, we have identified critical limitations in previous work: (1) General LLIE approaches have not overcome issues of poor generalization on nighttime driving datasets: the inferred images exhibit artifacts, noise, and severe color distortions. (2) Only a few DL-based Low-Light Image Enhancement (LLIE) studies have considered deployment costs (computational resources, inference time), which is not conducive to edge resource deployment. In light of these observations, this paper proposes a specialized lightweight enhancement network for night-time scenarios in intelligent driving, named VELIE. This network demonstrates robust generalization capabilities across multiple synthetic and real datasets, while also offering better interpretability, enhanced global semantic information augmentation, and faster inference times.

## 2. Literature Review

### 2.1. Traditional Methods

The endeavor to enhance images captured in low-light conditions, known as Low-Light Image Enhancement (LLIE), is a topic that has garnered considerable attention within the realm of digital image processing. Initial approaches to this challenge predominantly hinged on principles derived from signal processing and optical theories, forming the foundation of early methodologies in this field. Early methods were generally based on signal processing and optical theories. Histogram equalization [11] and its derivatives [12,13,14] improve image contrast and brightness by globally or locally reallocating pixel brightness levels. Methods like gamma correction and log correction are based on intensity transformations, modifying the mathematical representation of each pixel in the original image’s grayscale to achieve the desired illumination output. In addition to computational approaches, another category of LLIE methods is founded on the Retinex theory [15], which hypothesizes that a color image can be decomposed into reflectance and illumination components by simulating the human eye’s imaging principle. In the Retinex-based LLIE approaches, Single-Scale Retinex (SSR) [16] uses a single-scale Gaussian filter to compensate for the illumination map. To address the color distortion or excessive contrast enhancement in SSR, Multi-Scale Retinex (MSR) [17] was proposed, employing multi-scale Gaussian filters and introducing a color restoration factor for more natural color correction. Another traditional LLIE method type is transformation-based methods, such as wavelet transforms, gamma correction, and logarithmic transformations [18,19,20]. The core idea of transformation-based methods is to perform nonlinear transformations on the local or global contrast of an image, thereby achieving signal enhancement in the spatial or frequency domain.

One major advantage of traditional methods is their low computational cost and more deployment-friendly algorithmic environment, which is highly beneficial for on-board hardware deployment. Additionally, due to their foundation in signal processing and optical models, traditional methods possess good interpretability. In the processing of a low-light image by traditional methods, whether in the spatial or frequency domain, the transformation process of each pixel or digital signal can be mathematically modeled. This process is observable and tunable, and since each step is predefined, the sequence of execution and the effect of each operation is very clear. Despite their predictability and interpretability due to the typically non-complex nonlinear operations of traditional methods, these simple transformations also have numerous drawbacks. The degradation of illumination is inherently a nonlinear distortion process, with noise often arising in the reflectance component obtained from the low-light image according to advanced Retinex theory [21]. The core idea of traditional LLIE methods often involves amplifying the sensor’s gain, so the noise present in darkness and the imbalance of colors in the original data are also amplified. The loss of details in dark areas makes it challenging for traditional image processing methods, which lack the ability to learn contextual features, to replenish degraded detail information. Consequently, deep learning networks, with better capabilities for contextual semantic information learning and inference, have been proposed to address the limitations of traditional image processing methods.

### 2.2. Learning-Based Method

The domain of Machine Learning (ML) and Deep Learning (DL)-based LLIE tasks can be primarily categorized into two types. The first type encompasses learning networks based on traditional enhancement theories. RetinexNet [22] was a pioneering model that combined Retinex theory with deep learning networks, achieving learning-based image decoupling and low-light image enhancement. Subsequently, KinD and KinD++ [21,23] adopted a similar decoupling-before-enhancement approach, designing deep networks that allow manual adjustment of the output illumination level. URetinexNet [24] then established an unfolding Retinex theory to solve the optimization problem in the Retinex theory. Besides the Retinex theory, learning-based networks centered on differentiable histogram equalization [25], and those guided by wavelet transform [26] have also been proposed. Deep enhancement networks that integrate traditional low-light theories maintain interpretability while converting the manual parameter optimization process of illumination and detail enhancement into an automated update procedure. This process essentially moves away from manual trial-and-error and subjectivity, with the system learning the mapping relationships between lighting conditions and mathematical expressions from a vast amount of prior data.

The second type of LLIE method views low-light enhancement as a generative model-like task in deep learning. In these studies, low-light images are approximated as an entirely black mask, prompting the model to understand the underlying probability distribution of the samples (low light) to generate new data (synthetic normal light) that resembles real data (normal light). Gong [27] was the first to introduce the classic generative model, Generative Adversarial Network (GAN), into low-light enhancement, designing the first unsupervised low-light enhancement network. This model demonstrated excellent generalizability in real-world image enhancement [28]. Subsequently, LLFlow, based on a more powerful generative model, the flow model [29], was proposed, achieving high scores in image quality assessment metrics. With the outstanding performance of diffusion models across all generative models, networks based on diffusion generative model baselines for LLIE [30,31] have also been validated for their exceptional performance.

### 2.3. LLIE in Driving Scenarios

Being different from general LLIE applications, LLIE technology in intelligent driving systems serves as a critical safety feature. Its primary objective is not merely to provide clearer and more precise visual information but to aid the driving perception system in better understanding the surrounding environment. The essential requirements of LLIE tasks in driving scenarios are twofold: real-time computation and the enhancement of critical features. This means that the algorithm must restore features such as road signs, pedestrians, and vehicles in darkness to normal illumination levels in an extremely short time, enabling the autonomous driving system to recognize and react more accurately.

Historically, traditional LLIE methods like histogram equalization and gamma correction [11,18], due to their simple computational process and rapid execution, were suitable for scenarios demanding high real-time performance. However, they have not been widely adopted because they perform poorly in processing most low-light driving conditions, struggling with noise control and detail recovery in extremely dark areas and under multiple light sources. To address this, the authors of [32] proposed a method based on the MSR approach, introducing a more complex computation with Retinex-based techniques for better enhancement effects while controlling the processing time to 0.2 s. Similarly, the authors of [33] introduced a biologically-inspired traditional computation method, utilizing a retinal imaging theory similar to the Retinex theory to achieve more natural night-driving data enhancement, and achieving faster processing speeds by reducing image resolution.

While traditional Retinex-based models can provide better enhanced image quality, they still struggle with noise control and cannot achieve millisecond-level real-time computation. Therefore, Deep Learning (DL)-based LLIE methods are crucial, as DL-LLIE can maintain real-time processing speeds while effectively improving image quality, reducing noise, and restoring details. LE-Net [34] achieved efficient DL-based enhancement in driving scenarios and can be used in extremely dark conditions. AugGAN [35] and IA-GAN [36] have both acquired knowledge of simulated lighting conditions, using GAN models for all-day driving environment data simulation, making their enhancement networks adaptable to multiple detectors. Regarding efficient computation, according to the authors of [37], the current Swin Transformer-based methods can perform calculations with as low as 90k parameters and a computation time of 0.004 s, significantly lower than the processing time of traditional methods, thereby creating a faster response time for real-time hazard avoidance in vehicles.

## 3. Method

In view of the motivations discussed earlier, we propose a lightweight network, VE-LIE (Vehicle-Based Efficient Low-Light Image Enhancement), which exhibits excellent generalization on driving datasets. For any raw RGB input, according to the Retinex theory, it can be considered as a combination of reflected and ambient illumination. Our network employs a qualitative operation, enhancing the reflectance and illumination components independently in two separate branches, as illustrated in Figure 1. The Mask Decomposition Network (MDN), Reflectance Enhancement Transformer (RET), and Gamma U-Net (G-UNet) exist as key components of our main network and will be explained in detail in the following section. Our VELIE adopts a supervised learning approach in that paired datasets allow the network to effectively model the illumination degradation function. Although this approach slightly increases processing speed, it achieves significantly better results compared to mixed enhancement techniques.

Traditional indoor low-light supervised paired datasets do not address the complex lighting and details present in driving environments. The RGB images captured in intelligent driving often feature complex urban lighting and a variety of targets, which is markedly different from indoor scenes or low-light datasets with no light sources or only a single weak light source. Therefore, we chose CNN [38] and Vision Transformer [39] as different baselines to address the varying complexities of degradation in low and high-frequency information. The degradation on the Illumination Map is linear grayscale attenuation, which can be processed with a smaller parameter CNN. In contrast, degradation on the Reflectance Map includes dark area noise, blurriness, and unclear contextual information, necessitating the use of a more detail-oriented and powerful transformer for processing.

### 3.1. Mask Decomposition Network (MDN)

In the Retinex theory, a raw RGB input can be divided into two same-scale maps: a reflectance map and an illumination map. The reflectance component generally stores structural, texture, and color information, while the illumination component primarily contains pure lighting information and can be considered as a two-channel grayscale illumination map. For a low-light image, the degradation in dark areas is mixed, including low-frequency degradation such as lighting intensity, as well as high-frequency degradations like color distortion, noise, and texture blurring. This presents a challenge: direct enhancement on raw RGB can lead to uneven variable weights in the degradation function at each pixel, making it difficult to balance with a single convolution operation. This often results in local overexposure or under-enhancement, along with the loss of detail.

To circumvent this, some networks opt for a decoupling approach, breaking down the original raw RGB input into two maps for targeted brightness and detail enhancement. We employ a shallow CNN for the decoupling process, as displayed in Figure 1, following an encoder-decoder structure. Specifically, the input image first passes through successive convolutional layers, each with 3 × 3 filters followed by a ReLU [40] activation function. After certain convolutional layers, 2 × 2 max pooling is performed to reduce the size of the feature maps, thus constructing the down-sampling path of the encoder. In the decoder, the network gradually restores the size of the feature maps through convolutional layers and ReLU activations, combined with up-sampling operations. During this process, feature maps from the encoder are directly copied to the corresponding layers in the decoder to preserve and restore spatial information. The network ultimately outputs two components: the reflectance map and the illumination map. These two components are then independently enhanced by two branches, the Reflectance Enhancement Transformer (RET) and the Gamma U-Net (G-UNet).

### 3.2. Reflectance Enhancement Transformer (RET)

In the foundational Retinex theory, the reflectance map is assumed to be invariant under different lighting conditions, as it reflects only the intrinsic properties of objects (such as texture and color information) and is not altered by lighting. However, as highlighted in KinD [21], noise often occurs in darker regions, and thus the reflectance map derived from low-light dark areas tends to be noisier. Consequently, the information in the reflectance map actually varies under different lighting conditions, making this nonlinear degradation more challenging to model than the degradation caused by lighting itself. As mentioned earlier, the environmental complexity in driving datasets is significant. Vehicles need to perceive not just other vehicles but also pedestrians, cityscapes, roads, road signs, traffic lights, etc. The restoration of varied texture and color information of these complex elements is crucial, as downstream tasks depend on this low-frequency information for semantic scene understanding. Therefore, we employ a transformer-based deep learning network architecture to enhance the lost details and color distortion in the reflectance map of low-light images, leveraging its superior global perception capabilities. Unlike CNNs, which focus on local perception, transformers can capture the entire image’s global dependencies through their self-attention mechanism. This means that transformers can consider the interconnectedness of information across various local patches in an image when processing detailed information. This is particularly vital for processing the reflectance map in low-light driving conditions, where the loss of information due to insufficient lighting necessitates broader contextual support to restore details and ensure the harmony of structure, texture, and color of the same object. Additionally, since the transformer structure is not limited by a local receptive field, it can more effectively model long-distance dependencies within the image.

However, the enhanced performance of transformers comes at the cost of high computational demands. Considering the need for real-time inference in driving scenarios, we have chosen the Swin Transformer [41] structure as our backbone. The Swin Transformer introduces a shifting window mechanism for computing self-attention on the foundational transformer architecture. This mechanism allows the model to perform self-attention calculations within local windows while connecting the entire image through the shifting of these windows. Traditional transformers, in contrast, compute self-attention considering the whole image or sequence, which can lead to excessive computational loads. In addition to its lower computational resource requirements, the shifting window mechanism in the Swin Transformer also offers greater flexibility in processing different image areas, facilitating smooth enhancement in specific regions (such as shadows or highlights), thus preventing artifacts or overexposure.

In our Reflectance Enhancement Transformer (RET) branch, which is depicted in Figure 2, we employed a Swin Transformer structure based on SwinIR [42] as the baseline. This structure primarily consists of four Swin Transformer Components (STCs), each comprising four Swin Transformer Layers (STLs). We judiciously reduced the number of STCs and STLs from the original SwinIR baseline, as our experiments indicated that excessive modules could lead to overfitting. Each residual STL is mainly composed of two parts: a Multi-head Self-Attention (MSA) module and a Multi-Layer Perceptron (MLP). The MSA module specifically employs a shifting window strategy, the so-called Shifted Window MSA, which facilitates the shifting of window positions between consecutive STCs. All the modules mentioned above are displayed in abbreviated form in Figure 2, which also clearly depicts the connections between them. This strategy aids in establishing connections between windows, thereby integrating global information. In the original SwinIR baseline, both MSA and MLP are preceded by a Layer Normalization (LayerNorm) operation. However, LayerNorm segments the computation of the self-attention mechanism into pixel-level operations, reducing the network’s inference speed. Therefore, in our study, Batch Normalization (BatchNorm) is used to replace LayerNorm [43]. In the inference phase, BatchNorm merges with the convolution operation, avoiding additional computational time, which is crucial for real-time enhancement in driving edge cases. The outputs of each MSA and MLP module are added back to their inputs through residual connections to prevent the vanishing gradient problem in deep network training.

After enhancement through RET, to avoid biased errors in the color enhancement of the reflectance due to a lack of constraints during training, we introduced a custom-designed color loss, named the Color Histogram Matching Loss ($L_{\mathrm{hist}\;}$). This loss aims to ensure that the enhanced image’s color distribution closely resembles that of the original image. The loss function achieves this by comparing the color histograms of the two images across each color channel. Since we chose the RGB color space to represent the reflectance map, it is necessary to consider the red (R), green (G), and blue (B) channels separately.

Firstly, for each color channel, the color histogram of the image needs to be calculated before and after enhancement. Assuming our image is (H×W) pixel size, with color values for each pixel ranging from 0 to 255, for each color channel, we divide the color values into (B) buckets, each representing a certain range of color values. The color histogram Hist can be calculated in the following way:(1)$${Hist}_{c}(i)=\frac{1}{H\times W}\sum_{x=1}^{H}\sum_{y=1}^{W}1_{\left\{bin(I_{c}(x,y))=i\right\}}$$
where c represents the color channel (R,G,B), $I_{c}(x,y)$ is the color value of the channel in the image at coordinates (x,y), $bin(I_{c}(x,y))$ is the bucket number to which the color value belongs, and 1 is the indicator function. When its internal condition is true, the value is 1, otherwise, it is 0. ${Hist}_{c}(i)$ is the normalized histogram value of the ith bucket on channel c.

Next, we use the Kullback–Leibler (KL) [44] divergence to measure the difference between these two histograms. The KL divergence is a measurement method used to measure the difference between two probability distributions, which directly optimizes the difference in color distribution to help the enhanced image visually maintain color consistency with the original image. For each color channel, the KL divergence is defined as follows:(2)$$D_{KL}\left({\;\mathrm{Hist}\;}_{\mathrm{enhanced}\;}^{c}∥{\;\mathrm{Hist}\;}_{\mathrm{original}\;}^{c}\right)=\sum_{i=1}^{B}{\;\mathrm{Hist}\;}_{\mathrm{enhanced}\;}^{c}\left(i\right)\log\left(\frac{{\mathrm{Hist}}_{\mathrm{enhanced}\;}^{c}\left(i\right)}{{\;\mathrm{Hist}\;}_{\mathrm{original}\;}^{c}\left(i\right)}\right)$$

Here, ${\;\mathrm{Hist}\;}_{\mathrm{enhanced}\;}^{c}$ and ${\;\mathrm{Hist}\;}_{\mathrm{original}\;}^{c}$ are the color histograms of the enhanced and original images on channel c, respectively. Finally, the color histogram matching loss is the sum of the KL divergence on all color channels:(3)$$L_{\mathrm{hist}\;}=\sum_{c\in R,G,B}D_{KL}\left({\;\mathrm{Hist}\;}_{\mathrm{enhanced}\;}^{c}∥{\;\mathrm{Hist}\;}_{\mathrm{original}\;}^{c}\right)$$

### 3.3. Gamma U-Net (G-UNet)

When the image exposure is insufficient or excessive, the gamma transform can adjust the contrast of the image [19]. Through different approaches, the γ value is used to enhance low or high grayscale parts. Through this nonlinear transformation, the gamma transform can expand the low or high grayscale range of the image, thereby correcting darker or brighter images. The gamma transform follows the following formula for operation:(4)$$I_{\mathrm{out}\;}=I_{\mathrm{in}\;}^{\gamma }$$

In practical applications, the gamma transformation is applied independently to each pixel value in an image. Unlike histogram equalization or linear transformations, it operates on a pixel-wise scale. Such pixel-wise processing grants the algorithm enhanced flexibility in augmenting various image contents. It can adapt to the specific needs of different regions within an image, effectively adjusting both dark and bright areas.

In operation, selecting different gamma γ values allows for the enhancement or reduction in the contrast at each pixel. For instance, a gamma value less than 1 (γ < 1) is typically used to brighten low-light images, whereas a gamma value larger than 1 (γ > 1) is utilized to dim overly bright or overexposed images.

Traditionally, the appropriate gamma parameter is manually adjusted to balance the enhancement between noise and illumination. However, this process is unreliable, as the enhancement effect is dependent on the operator’s intuitive assessment of image quality. Since there are no standard enhancement results for low-light enhancement, image quality metrics like the Peak Signal-to-Noise Ratio (PSNR), Structural Similarity Index Measure (SSIM) [45], and Natural Image Quality Evaluator (NIQE) [46] are often used to evaluate the effectiveness of enhancement. The manual tuning of gamma parameters, relying solely on subjective user studies, makes achieving optimal metric scores and visual quality uncontrolled. Moreover, in the context of autonomous and assisted driving, real-world lighting conditions are variable and complex. Using fixed parameters for enhancement cannot avoid issues like overexposure, insufficient enhancement, or localized distortion.

Therefore, in our model, we propose a learning-based method to automatically regularize the gamma parameter on the illumination map. In this automatic regularization process, the gamma parameter is adaptively obtained through propagation, loss computation, and parameter update algorithms. The rationale behind using a learning-based method for parameter update lies not just in automation and efficiency but also in the superiority of representation learning in simulating complex lighting environments. Convolutional layers learn high-level feature representations through stacked nonlinear transformations, an automatic feature extraction mechanism that allows the model to capture more complex and abstract data patterns.

In our approach, a supervised learning model based on loss minimization is ideally suited for updating the Gamma parameter. Beyond learning basic lighting and core semantic features of driving scenes, we aim for the network to learn underlying distributions and patterns, achieved by maximizing posterior probability in a Bayesian framework and minimizing the empirical risk in a frequentist framework. Thus, selecting an appropriate network architecture and embedding the Gamma learning layer at the right position is critical. Considering that gamma parameter adjustment involves nonlinear mapping, the chosen network architecture should be capable of handling high-dimensional and nonlinear feature spaces. It should also capture both first and second-order statistical features of driving illumination, understanding and integrating higher-order statistical features.

The specific implementation location of the gamma transformation also warrants consideration. Rather than being just a preprocessing step in a single block, gamma transformation can be integrated as a trainable layer within certain stages of a deep learning network. This approach aims to enhance the network’s adaptability and performance, especially in response to varying lighting conditions for each data point in the dataset, allowing the network to autonomously learn and optimize gamma values during training. In our model, the gamma transformation layer is embedded within both the encoder and decoder. This layer operates based on a parameterized function, with the gamma value optimized as a trainable parameter. Specifically, the layer operates according to Formula (5):(5)$$O(i,j)=I(i,j{)}^{\gamma }$$
where O(i,j) is the brightness value of the output image at position (i,j), I(i,j) is the brightness value of the input image at the same position, and γ is the parameter of the Gamma transform, which controls the degree of nonlinearity in brightness adjustment.

Given that gamma transformation operates on a pixel-wise level, selecting a learning network baseline with a matching receptive field is crucial. In traditional gamma transformation, pixel-level enhancement essentially only considers pixel-level fine-tuning without accounting for the understanding of contextual information between pixels. Therefore, in considering a learning-based approach for gamma transformation, it is possible not only to utilize ground truth comparisons to fit precise gamma parameters but also to control the coordination of each pixel based on input information within a small neighborhood around it. Consequently, U-Net [47] has been chosen as the baseline for integrating the gamma layer, due to its exceptional feature extraction and fine pixel-level processing capabilities. Originally proposed for semantic segmentation tasks, U-Net, with its encoder-decoder structure and skip connections, has also been proven to be an excellent baseline for image enhancement tasks [48,49,50]. Most importantly, U-Net is relatively efficient and compact, with a reasonable number of parameters and inference speed, making it suitable for deployment in edge computing architectures at the driving end.

In the traditional U-Net architecture, which is shown in Figure 3, the decoder consists of several upsampling operations connected by transposed convolutional layers and convolutional layers, aiming to gradually restore image resolution and detail. In our network design, we chose to insert the Gamma Layer (GL) after each upsampling operation. This allows the network to adapt to different lighting conditions earlier when processing higher-resolution feature maps, thus effectively improving global brightness rather than local details. Moreover, we avoid adding gamma transformation modules after every convolution or skip connection to reduce the complexity and training difficulty of the network in adjusting the brightness at multiple levels.

Since we set the gamma parameter as a learnable variable during network training, this learning process is determined through backpropagation and gradient descent. Initially, the gamma parameter γ is initialized to a preset value of 1, as 1 in a gamma transformation indicates that the parameter will not alter the image’s brightness. Then, during training, the input image undergoes forward propagation through the network, and as the image passes through the gamma transformation layer, each pixel value is adjusted according to Formula (5). The network’s adjusted output is then compared with the desired output, i.e., the ground truth, to calculate the loss function. Apart from the original U-Net loss function $L_{1\;}$ [51], we designed a gamma transformation-based loss function $L_{\mathrm{Gamma}\;}$ to control the disparity between the network output and real labels, also regulating the range of gamma values through a regularization term.

In the loss function tailored for G-UNet, a weighted squared difference and gamma value range constraint scheme for optimization is adopted, making it more precise and effective. Firstly, we set the weight proportional to the degree of deviation of the gamma value from the target value using the following formula. This way, the greater the deviation of the gamma value from the target, the larger the penalty. The optimization formula can be expressed as follows:(6)$$\mathrm{Optimized}\;\mathrm{Weighted}\;\mathrm{Squared}\;\mathrm{Deviation}=\lambda \sum_{i=1}^{n}\left|\gamma_{i}-\mu \right|{\left(\gamma_{i}-\mu \right)}^{2}$$

To ensure that the gamma value is within a more reasonable and flexible range, a smooth penalty function is introduced, such as using a logistic function instead of a hard maximum function. This can reduce mutations and provide smoother gradients. The formula is represented as follows:(7)$$\begin{array}{c} Improved\;Range\;Constraint=\kappa \sum_{i=1}^{n}\left[\frac{1}{1+e^{-a\left(\gamma_{i}-\gamma_{max}\right)}}+\frac{1}{1+e^{a\left(\gamma_{i}-\gamma_{min}\right)}}\right] \end{array}$$

The basic loss function, the optimized weighted squared difference, and the improved gamma range constraints are combined to form the total loss function. The formula can be expressed as follows:(8)$$L_{\mathrm{Gamma}\;}=\lambda \sum_{i=1}^{n}\left|\gamma_{i}-\mu \right|{\left(\gamma_{i}-\mu \right)}^{2}+\kappa \sum_{i=1}^{n}\left[\frac{1}{1+e^{-a\left(\gamma_{i}-\gamma_{max}\right)}}+\frac{1}{1+e^{a\left(\gamma_{i}-\gamma_{min}\right)}}\right]$$

In Formulas (6)–(8), λ and κ are weight coefficients to balance the loss from different parts; $\gamma_{i}$ is the gamma value of the ith pixel; μ is the target gamma value; $\gamma_{max}$ and $\gamma_{min}$ are the maximum and minimum permissible ranges for the gamma value, respectively; and a is a parameter to adjust the slope of the logistic function, controlling the strictness of the range constraint. This loss function design aims to achieve more detailed and efficient optimization in the image enhancement process by applying weighted penalties to gamma value deviations and flexible range constraints on the gamma values. After calculating the $L_{\mathrm{Gamma}\;}$ loss function, the result of the loss is used for the backpropagation process. In this process, the contribution or gradient of each parameter (including gamma parameter γ) to the loss is calculated using a chain rule to update the gamma parameters. We use the gradient descent variant Adam optimizer to update the gamma parameters. In this update step, the loss function will gradually be reduced; that is, adjusting the gamma value gradually makes the enhanced low-light result close to the ground truth. Throughout the training phase, the network will repeat the aforementioned processes of forward propagation, loss calculation, backpropagation, and parameter updates until the network fits the stopping condition. The optimized gamma parameters will be saved and used in the testing phase.

### 3.4. Feature Fusion Output Layer

After an input image has been enhanced through the RET and G-UNet branches, it is essential to recombine these two separate components into the final RGB output image. This recombination process is conducted in accordance with the principles of the Retinex theory, where an RGB image is perceived as the point-wise product of the reflectance map and the illumination map. Consequently, the enhanced reflectance and illumination maps can be merged using the same multiplication operation.

## 4. Experiment

### 4.1. Dataset

Similar to most LLIE tasks, we have selected the LOL dataset [22] as a part of the training dataset. The LOL dataset consists of 485 paired training images and 15 test images with the ground truth, collected under stepwise illumination conditions in indoor scenarios. Not only does the LOL dataset simulate varying degrees of low-light scenarios but the chosen scenes also contain rich color and structural information, making it suitable as a5high-quality training set. However, the dataset collection scenes are almost exclusively indoors, lacking the complexity of lighting in driving scenarios and missing semantic information on core driving-related targets. Solely training with the LOL dataset possibly diminishes the effectiveness of the network in restoring unique features in driving scene images. In order to generalize the network’s inference in driving scenarios, we have collected our VELIE dataset to enrich the training and testing scenes.

The VELIE dataset is divided into two parts: paired (P-VELIE) and unpaired (UP-VELIE). In the P-VELIE portion, we adopted the method from RENOIR [52], using a fixed Nikon Z8 camera (Nikon Corporation, Tokyo, Japan) to capture the same scene with different apertures, designating the normal lighting images as the ground truth and the others as low-light scenarios. This fixed shooting mode allowed us to collect 500 pairs of driving scene low-light datasets, which were split into 485 training and 15 testing pairs. These pairs were oriented toward driving scenarios and aligned pixel-by-pixel into supervised pairs using the three-step method from RENOIR. In the UP-VELIE part, we used an SG2-AR0231C-0202-GMSL vehicle-mounted RGB camera for real-world road scene capture in Shanghai and Suzhou, China, and London and Liverpool, United Kingdom. The dataset ultimately comprised 1000 frames, from which we selected 200 high-quality images as the UP-VELIE test set.

Thus, in the training phase of our network, we first used the LOL dataset’s 485 training images to compare baseline performance on general LLIE tasks, followed by mixed training with the 485 P-VELIE training images and the LOL training set to validate the different baselines’ generalizability to driving scenes. In the testing phase, we initially employed the LOL dataset’s 15 test images to assess the training effectiveness using only the LOL dataset, with the results presented in Figure 4 and Table 1. The mixed metric results of the 15 LOL test images and 15 P-VELIE dataset images are also shown in Table 1. Since each test image in the LOL and P-VELIE datasets has the ground truth, we employed common reference-based metrics like PSNR and SSIM to evaluate the quality of the enhanced images. To verify our network’s effectiveness in more realistic scenarios, we conducted tests on the UP-VELIE. Moreover, to demonstrate the excellent generalizability of the VELIE enhancement network, we also conducted tests on the real-world dataset LIME [53] and the driving-based dataset DarkZurich [54]. Visual results on the above three test sets are displayed in Figure 5. We calculated the average No-Reference Image Quality Evaluation (NIQE) scores and average inference time for a single image on a random selection of 100 Images from both the DarkZurich, UP-VELIE, and LIME datasets, with the results presented in Table 2. All of the learning-based baselines we compared were ensured to have the same dataset configuration as ours, allowing for a comparative analysis of the superiority among different enhancement method baselines.

### 4.2. Implementation

In our evaluative study, we strictly followed the training configurations as described in the source documents of the methodologies we assessed, ensuring the integrity of our experiments by performing both the training and testing phases within the identical GPU framework. All deep learning (DL)-based approaches underwent training and testing solely on GPU platforms. The deployment of conventional computational methods for low-light enhancement was also executed on the same GPU, with all performance metrics being analyzed via GPU processing. Regarding the training duration, our model underwent an 11 h training period on a 4090 GPU using PyTorch,(PyTorch v2.1.2) with detailed training specifications tailored for this GPU environment: the input images were resized to a resolution of 600 × 400 for the training process, with the total number of epochs set at 200 and an initial learning rate of 1 × 10^−4^. Additionally, the batch size was determined to be 16. For the Reflectance Enhancement Transformer (RET) branch specifically, we utilized a patch size of 32 × 32.

Highlighting the economic considerations and deployment complexities of edge computing, it is important to note that theoretically, all DL-based checkpoint inferences could be made compatible with CPU usage. The application of CPU-based computations for traditional methods is also viable. Nonetheless, the transition to CPU processing could lead to a substantial increase in computation times, making the use of onboard GPUs a preferable choice under current technological and practical considerations. Looking forward, investigating the integration of VELIE within automotive edge computing frameworks and the execution of empirical nocturnal driving tests emerges as a critical avenue for future exploration, aiming to substantiate the utility and efficacy of the proposed enhancement framework in real-world scenarios.

### 4.3. Enhancement Results

The distinctions among the performance metrics of the various illumination enhancement techniques highlighted in the supplied Figure 4 and Figure 5 and Table 1 and Table 2 are pronounced. Our selection of baselines for comparative analysis is predicated on a multifaceted approach to the issue of low-light enhancement, encapsulating traditional methodologies CLAHE [14] and MSRCR [17], DL-based techniques RetinexNet [22], KinD++ [23], and URetinexNet [24] grounded in the Retinex theory, the prior information-based method PairLIE [55], and generative models EnGAN [28] and LLFlow [29], all of which represent the pinnacle of the state-of-the-art in their respective domains. The intent is to assess their efficacy across distinct datasets.

CLAHE, as an archetypical conventional image enhancement strategy, endeavors to ameliorate visual acuity by modulating local contrast within the image. However, its performance, as quantified by the PSNR and SSIM metrics on the LOL dataset, yields values of 16.21 and 0.57, respectively, with a marginal uptick observed on the P-VELIE dataset to 17.76 and 0.63. This increment betrays the inherent limitations of CLAHE in preserving the structural integrity of images, particularly within scenarios characterized by extensive dynamic ranges. Visual inspection of Figure 4a corroborates that, while contrast enhancement is achieved, CLAHE is predisposed to generating visually discordant effects in regions suffering from uneven illumination or overexposure, thereby compromising detail fidelity.

In a similar vein, MSRCR, another traditionalist technique with a focus on color restitution and contrast augmentation, demonstrates suboptimal PSNR and SSIM values across both datasets, with its deficiencies becoming conspicuously egregious on the LOL dataset. The resultant imagery post-MSRCR enhancement, as depicted in Figure 4b, shows a lackluster improvement in luminosity and a marked degradation in color veracity.

Conversely, deep learning methodologies like RetinexNet and KinD++—both derivatives of the Retinex theory—demonstrate a superior command in image quality reconstruction, as evidenced by their PSNR and SSIM indices. Notably, KinD++ attains a PSNR of 21.30 on the LOL dataset, underscoring its robustness in image quality enhancement. Yet, juxtaposed with VELIE, both Retinex-based methods exhibit discernible deficits, not only in terms of visual aesthetics as illustrated in Figure 4c,d but also in terms of quantitative benchmarks. This discrepancy is attributable to the methods’ inability to restore verisimilitude in detailing and coloration under extreme lighting conditions.

Generative models such as EnGAN and LLFlow, employing the generative adversarial network framework, aspire to synthesize images of superior quality through the extrapolation of extensive data corpuses. Their performance, denoted by the PSNR and SSIM metrics, is commendable on both test sets, with LLFlow’s SSIM reaching a notable 0.91 on the LOL dataset, indicating their proficiency in structural information preservation. Nonetheless, the visual samples in Figure 4e,f delineate that despite commendable detail and texture recovery in certain regions, these models grapple with maintaining color accuracy, lighting uniformity, and the overall naturalness of the imagery, which could potentially undermine tasks that hinge on precise color discrimination, such as the identification of traffic lights and road markers.

In summary, while the aforementioned methods each exhibit unique strengths, they collectively falter when paralleled against the VELIE approach, both in quantitative image quality assessments and qualitative visual outcomes. VELIE not only registers superior PSNR and SSIM metrics on both test sets but also visually outperforms the competing methods in luminance, contrast, saturation, and detail preservation, as palpably demonstrated in Figure 4g. This is particularly salient in the context of processing images beset with complex lighting and expansive dynamic ranges, a feat made possible by the tailored Color Histogram Matching Loss ($L_{\mathrm{hist}\;}$) and the adaptive Gamma Layer (GL) illumination adjustment mechanism intrinsic to VELIE.

The incorporation of the inference time in the analysis, as elucidated by Table 2 and derived from real-world dataset evaluations, introduces a pivotal dimension to the discourse, particularly within the ambit of autonomous driving, where it directly correlates with the alacrity and real-time decision-making process of the system. In the exigent milieu of real-time driving scenarios, the imperative for image enhancement algorithms transcends mere excellence in visual quality; it mandates the expeditious processing of incoming image data to uphold safety and efficiency paradigms.

For conventional image enhancement techniques such as CLAHE and MSRCR, their performance deficits, already evident in experimental datasets, become conspicuously pronounced in real-world datasets. The tabulated data reveal that, while CLAHE boasts a relatively brief inference time of 0.47 s, surpassing the majority of deep learning counterparts, its performance, as gauged by the NIQE, falls short of the benchmark established by VELIE.

The augmentation in visual quality shown in Figure 5 offered by Retinex theory-based deep learning methods and generative models is tempered by the escalation of inference times, thereby positing a significant constraint. For instance, the inference durations for RetinexNet and KinD++ are clocked at 0.84 s and 0.41 s, respectively. Generative models like EnGAN and LLFlow, despite their commendable visual quality outcomes, are encumbered by LLFlow’s protracted inference time of 2.29 s—a duration untenable within the purview of autonomous driving applications.

In contradistinction, VELIE not only garners the lowest NIQE scores on real-world datasets, indicative of an image quality that most closely approximates the natural state, but it also distinguishes itself with an inference time of a mere 0.19 s, substantially outstripping all alternative methodologies. This facet is quintessential for autonomous driving, ensuring that the system can process images swiftly and accurately even under low illumination conditions, thereby enhancing the comprehension of and response to the driving environment. Such a symbiosis of speed and quality underscores the pivotal suitability of VELIE for driving scenarios, presenting a solution that safeguards security without compromising on response velocity.

### 4.4. Impact on High-Level Perception

To investigate the practical effectiveness of enhancement results in high-dimensional perception tasks in driving scenarios, we utilized an object detection network to explore the impact of results generated by different enhancement models on detection performance. In our original UP-VELIE dataset, a total of 2556 vehicle appearances were recorded across 1000 frames. We computed the accuracy of the same detection network on various enhancement outcomes, as illustrated in Table 3. Additionally, we calculated the average processing time, which encompasses both the time for image enhancement and detection inference.

Table 3 indicates that high-quality enhancement results indeed provide clearer visual information, enabling the object detection algorithm to more accurately identify vehicles on the road. Images processed with the VELIE method, due to their superior restoration of image details and colors in real datasets, minimizing noise and distortion, offer a better data foundation for the object detection algorithm. Conversely, lower-quality enhancement results, such as those from MSRCR and RetinexNet, may degrade the performance of the detection system. As shown in Figure 5, in extremely dark conditions, most models introduce additional artifacts. Such images interfere with the object detection algorithm, leading to erroneous target recognition—mistaking noise for targets or failing to identify real targets due to a lack of detail. In high-speed driving scenarios, even small errors can have serious consequences.

Furthermore, inference time is a critical factor. Rapid image processing capabilities ensure the real-time updating of visual information in dynamic environments, providing the latest data for the object detection algorithm. If the enhancement algorithm is slow, it may cause the detection system to react based on outdated information, which is unacceptable in rapidly changing driving environments.

By comparing different image enhancement methods in dark environments on vehicle target detection, the superior performance of the VELIE method is evident. In terms of detection rate, VELIE leads with a score of 94.3%, outperforming the second-ranked EnGAN by 2.2%. In processing time, VELIE is also impressive, requiring only 0.31 s for a single image to be processed, approximately 0.05 s faster than the second-place MSRCR, with 0.46 s. This significant time advantage is particularly important in applications such as autonomous driving systems, where high real-time demands are crucial. VELIE’s ability to ensure extremely high detection rates while significantly reducing processing time demonstrates its leading position in the field of nighttime driving target detection. This balance of significantly enhancing target detection accuracy while meeting real-time processing needs represents a significant advancement for the practical deployment of time-sensitive applications such as automated driving assistance systems.

## 5. Conclusions and Future Work

In this manuscript, we delve into a comprehensive analysis of RGB camera sensors, which are prevalently employed in intelligent vehicle perception systems. Our research culminates in the development of a novel deep learning-based enhancement framework, designated as the Vehicle-based Efficient Low-light Image Enhancement (VELIE). This framework integrates the Swin Vision Transformer with a gamma transformation augmented U-Net architecture, delivering unparalleled performance across a spectrum of driving datasets and markedly bolstering high-dimensional environmental perception under low-light conditions. VELIE distinguishes itself by achieving exemplary scores in PSNR, SSIM, and NIQE across diverse test sets. Visually, it supersedes competing methodologies in aspects such as luminance, contrast, saturation, and detail retention. A salient feature of VELIE is its rapid inference time, clocking at merely 0.19 s, which significantly outpaces other existing approaches. This rapid processing capability is pivotal in autonomous driving applications, ensuring efficient and precise image analysis even in scenarios of diminished illumination, thereby augmenting comprehension and interaction with the driving environment. The synergistic integration of expeditious processing and high-quality output underscores the indispensable suitability of VELIE in automotive contexts, offering a solution that concurrently prioritizes safety and response agility. The potential amalgamation of VELIE with an array of advanced high-dimensional driving perception tasks—encompassing semantic segmentation, depth estimation, 3D reconstruction, and object tracking—presents a fertile ground for further exploration and development in this domain.

Nonetheless, VELIE currently does not achieve inference times in the millisecond range. This limitation is attributed to the intricate computational demands of the transformer architecture and the presence of multiple sub-networks within the VELIE framework. Future endeavors should pivot toward exploring methodologies to expedite the inference process. Moreover, VELIE neither accounts for factors such as the image acquisition timing and the blurring effects caused by different weather conditions and vehicle speeds nor delves into deployment experiments across various industrial and driving cameras. These aspects represent critical variables that can significantly influence the performance and applicability of our framework. Future research should aim to address these gaps by conducting extensive testing under varied environmental conditions and speeds as well as by exploring the framework’s adaptability and efficiency across different camera systems with different shutter types. Such investigations will not only enhance the robustness of VELIE but also extend its applicability, ensuring its effectiveness in a broader range of real-world driving scenarios.

## Figures and Tables

**Figure 1 sensors-24-01345-f001:**
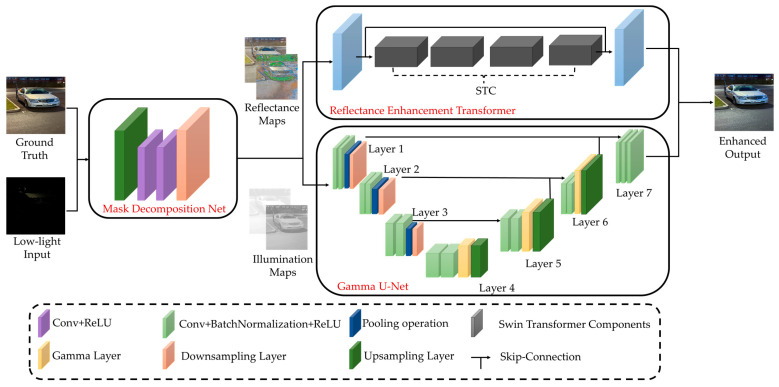
The general pipeline of the proposed Vehicle-based Efficient Low-light Image Enhancement (VELIE) Network.

**Figure 2 sensors-24-01345-f002:**
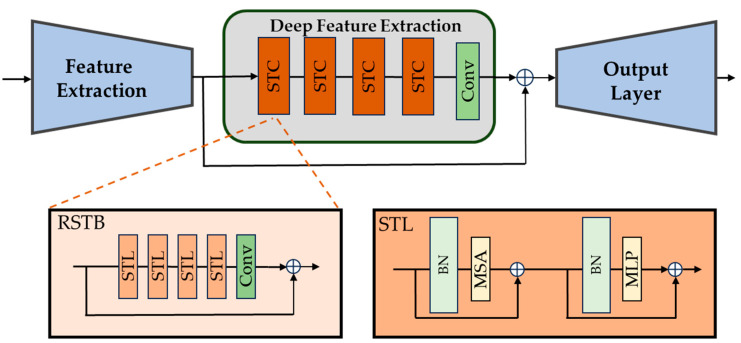
The detailed structure of the Reflectance Enhancement Transformer (RET).

**Figure 3 sensors-24-01345-f003:**
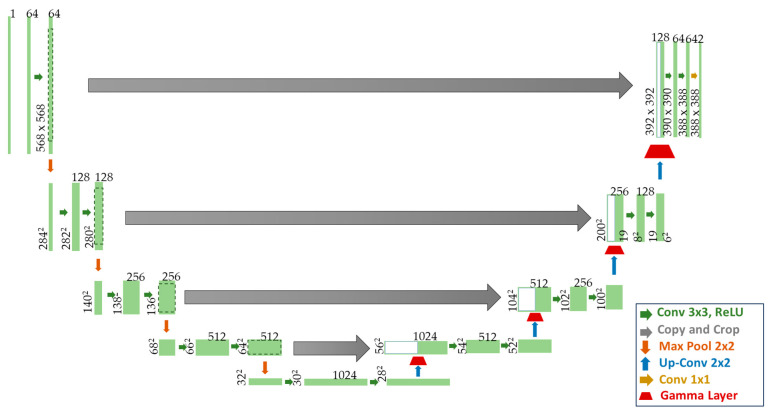
The detailed structure of Gamma U-Net (G-UNet).

**Figure 4 sensors-24-01345-f004:**
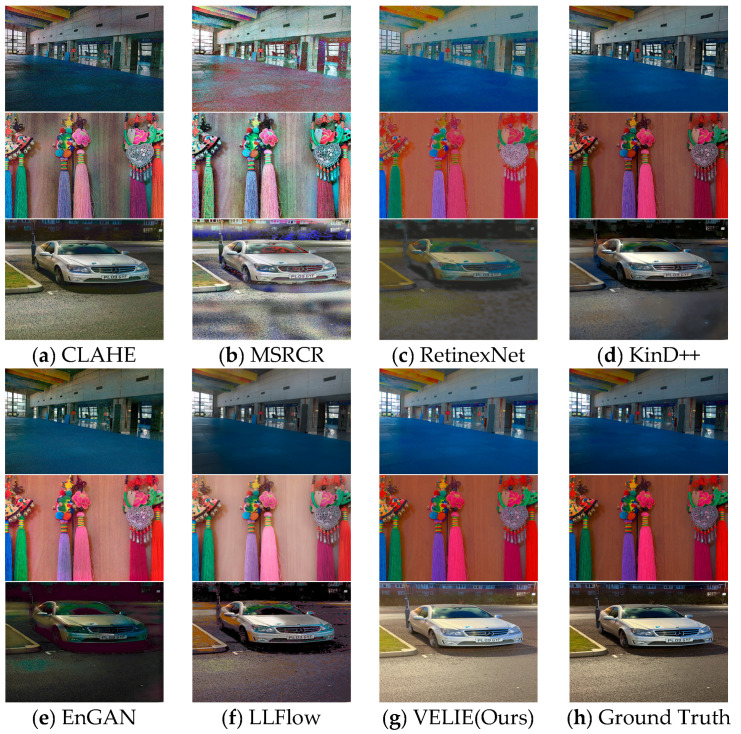
Visualization Comparison of VELIE and different baselines on LOL test set [22] and P-VELIE.

**Figure 5 sensors-24-01345-f005:**
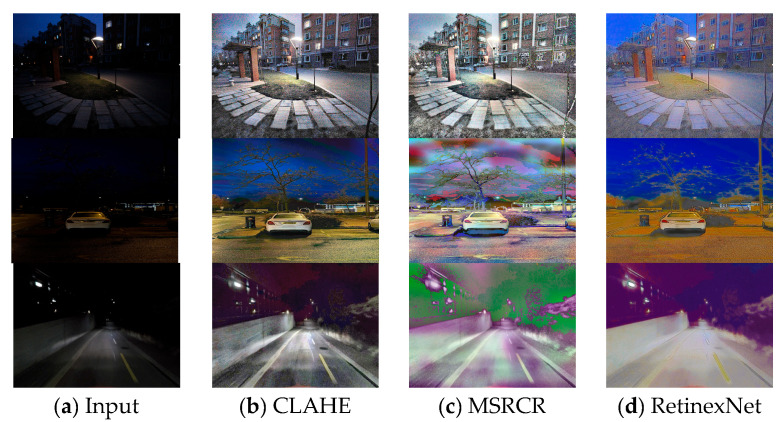
Visualization Comparison of VELIE and different baselines on LIME [53], DarkZurich [54], and UP-VELIE.

**Table 1 sensors-24-01345-t001:** Image quality metric comparison on LOL and P-VELIE from different methods.

Method	LOL Test Set		P-VELIE Test Set	
	PSNR ^1^	SSIM	PSNR	SSIM
CLAHE	16.21	0.57	17.76	0.63
MSRCR	11.24	0.35	9.97	0.24
RetinexNet	16.82	0.43	12.44	0.39
KinD++	21.30	0.82	21.07	0.79
EnGAN	17.48	0.65	19.95	0.70
LLFlow	25.72	0.91	20.34	0.83
PairLIE	19.51	0.74	17.72	0.69
URetinexNet	21.33	0.83	20.19	0.75
VELIE (Ours)	26.07	0.91	23.61	0.85

^1^ Higher PSNR and SSIM indicate better image quality.

**Table 2 sensors-24-01345-t002:** Image quality metric and inference time comparison on unpaired real-world dataset LIME, DarkZurich, and UP-VELIE from different methods.

Method	NIQE	Inference Time (s)
CLAHE	4.31	0.47
MSRCR	8.81	0.22
RetinexNet	7.25	0.84
KinD++	4.36	0.41
EnGAN	5.01	1.77
LLFlow	4.57	2.29
PairLIE	4.99	0.91
URetinexNet	4.33	1.12
VELIE (Ours)	3.97	0.19

**Table 3 sensors-24-01345-t003:** Detection rate and processing time comparison on DarkZurich and UP-VELIE from different methods.

Method	Detection Rate	Processing Time (s)
CLAHE	88.7	0.68
MSRCR	73.1	0.46
RetinexNet	77.5	1.02
KinD++	86.7	0.62
EnGAN	92.1	1.97
LLFlow	84.2	2.52
PairLIE	92.5	1.21
URetinexNet	92.4	1.33
VELIE (Ours)	94.3	0.41

## Data Availability

The new dataset used in this study, given its potential value in commercial applications, will not be made open-source at this time. We fully acknowledge the importance of open-source datasets for scientific research; however, considering business confidentiality and future commercial uses, we must retain exclusive rights to these data. We hope our readers can understand this decision and look forward to sharing our findings with the broader research community when appropriate.

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
