# Peer review of "VELIE: A Vehicle-Based Efficient Low-Light Image Enhancement Method for Intelligent Vehicles"

_sensors, 2024, doi:10.3390/s24041345_

Round 1
Reviewer 1 Report
Comments and Suggestions for Authors
The paper introduces "Vehicle-based Efficient Low-light Image Enhancement (VELIE)" as a solution to the limitations of RGB camera sensors in low-light conditions for Advanced Driving Assistance Systems (ADAS). VELIE, leveraging the Swin Vision Transformer and U-Net, claims state-of-the-art performance with a processing time of 0.19 seconds. The focus on cost-effectiveness and real-time processing is commendable. I found the reading experience enjoyable. The introduction is well-organized, and the literature review adequately addresses the missing key points. The material and methods section is thoroughly documented.
However, for further improvement, it would be beneficial to include a more extensive evaluation, potentially with comparisons to other state-of-the-art methods, to provide a clearer understanding of VELIE's competitive advantages. Additionally, insights into potential limitations or challenges in deploying VELIE in different real-world scenarios could enhance the paper's completeness.
Please provide explanations for abbreviations in formula 6 and 7.
Please address Figure 2 in the text while explaining the structure of RET.
Within the context of Table 2's processing time comparison, include details on the repetition time for each process and explicitly mention all datasets tested during this analysis.
For further comparative insights, consider addressing the CPU and GPU, providing a nuanced understanding of their respective roles within the study.
The study utilizes a common dataset to illustrate its effectiveness. Notably, the study overlooks considerations such as image acquisition time and blurring caused by the vehicle's velocity.
Consequently, for a real-time image processing application, it is crucial to assess the actual effectiveness when selecting an appropriate camera, whether it be a rolling shutter or global shutter. These aspects should be delved into in the results and conclusion sections to fortify the study's findings.
In short, the paper stands out for its innovative approach, practical focus, and efficient performance, making VELIE a promising contribution to the field of ADAS. With minor adjustments and additional considerations, it has the potential to become a significant reference in low-light image enhancement for driving assistance systems.
Reviewer 2 Report
Comments and Suggestions for Authors
This paper proposes a deep learning enhancement network named Vehicle-based Efficient Low-light Image Enhancement (VELIE), which utilizes Swin-Transformer combined with a gamma transformation integrated U-Net for decoupled enhancement of initial low-light inputs. Overall, the paper is well-structured, informative, and well-written, with sufficient experiments. However, there are some minor shortcomings:
1. It would be better to summarize the contributions of the paper in bullet points in the Introduction section. I think the third section of the paper, Motivation and Contribution, can be integrated into the Introduction section.
2. It would be better to provide more specific descriptions of the detailed parameters of the network and the specific details of the training, so that readers can easily reproduce the method of this paper.
3. It is hoped that the authors can further check and proofread the language expression and details of the paper.
Comments on the Quality of English Language
This paper proposes a deep learning enhancement network named Vehicle-based Efficient Low-light Image Enhancement (VELIE), which utilizes Swin-Transformer combined with a gamma transformation integrated U-Net for decoupled enhancement of initial low-light inputs. Overall, the paper is well-structured, informative, and well-written, with sufficient experiments. However, there are some minor shortcomings:
1. It would be better to summarize the contributions of the paper in bullet points in the Introduction section. I think the third section of the paper, Motivation and Contribution, can be integrated into the Introduction section.
2. It would be better to provide more specific descriptions of the detailed parameters of the network and the specific details of the training, so that readers can easily reproduce the method of this paper.
3. It is hoped that the authors can further check and proofread the language expression and details of the paper.
